# Reduced Graphene Oxide-Supported Iron-Cobalt Alloys as High-Performance Catalysts for Oxygen Reduction Reaction

**DOI:** 10.3390/nano13192735

**Published:** 2023-10-09

**Authors:** Jun Dong, Shanshan Wang, Peng Xi, Xinggao Zhang, Xinyu Zhu, Huining Wang, Taizhong Huang

**Affiliations:** 1School of Aerospace Engineering, Xi’an Jiaotong University, Xi’an 710049, China; dongjun@xjtu.edu.cn; 2Shandong Provincial Key Laboratory of Fluorine Chemistry and Chemical Materials, School of Chemistry and Chemical Engineering, University of Jinan, Jinan 250022, China; wss5680@163.com (S.W.); zhuxinyu112580@163.com (X.Z.); wanghuining1215@163.com (H.W.); 3Xi’an Modern Chemistry Research Institute, Xi’an 710065, China; xipeng-57@163.com; 4State Key Laboratory of NBC Protection for Civilian, Beijing 102205, China; xinggaozhang@aliyun.com

**Keywords:** oxygen reduction reaction, FeCo alloys, nitrogen-doped reduced graphene oxide, catalysts

## Abstract

Exploring non-precious metal-based catalysts for oxygen reduction reactions (ORR) as a substitute for precious metal catalysts has attracted great attention in recent times. In this paper, we report a general methodology for preparing nitrogen-doped reduced graphene oxide (N–rGO)-supported, FeCo alloy (FeCo@N–rGO)-based catalysts for ORR. The structure of the FeCo@N–rGO based catalysts is investigated using X-ray diffraction, scanning electron microscopy, X-ray photoelectron spectroscopy, and transition electron microscopy, etc. Results show that the FeCo alloy is supported by the rGO and carbon that derives from the organic ligand of Fe and Co ions. The eletrocatalytic performance is examined by cyclic voltammetry, linear scanning voltammetry, Tafel, electrochemical spectroscopy impedance, rotate disc electrode, and rotate ring disc electrode, etc. Results show that FeCo@N–rGO based catalysts exhibit an onset potential of 0.98 V (vs. RHE) and a half-wave potential of 0.93 V (vs. RHE). The excellent catalytic performance of FeCo@N–rGO is ascribed to its large surface area and the synergistic effect between FeCo alloy and N–rGO, which provides a large number of active sites and a sufficient surface area.

## 1. Introduction

Oxygen reduction reactions (ORR), as a major reaction of fuel cell cathodes, have been investigated for many years. The improvement of the catalytic activity of catalysts for ORR can greatly enhance energy-conversion efficiency, which is essential for the application of energy conversion devices [1]. At present, the Pt-group metal-based catalysts for ORR are widely adopted, despite the shortcomings of a high price and slow kinetics. In order to promote the commercial application of fuel cells, the exploration of low-cost, high-performance, long-term running-stability, non-precious metal-based catalysts has been studied [2,3,4]. However, as of now, there are still huge challenges for the substitution of Pt-based catalysts for large-scale application.

In recent years, high-performance, non-precious metal-based catalysts, especially transition metal-based catalysts, such as iron, cobalt, nickel-based catalysts, etc., have been widely investigated due to their abundant reserves in nature and low cost [5,6]. Theoretical studies showed that the 3D orbitals of transition metals could accommodate foreign electrons, thereby reducing the bonding strength of the intermediates of oxygen molecules such as OOH*, O*/OH* [7]. In theory, transition metal-based catalysts have great potential to be high-performance catalysts for oxygen reduction and other energy storage [8,9]. In particular, when these metals were integrated into alloys, such as FeCo, NiCo, and FeNi, they exhibited excellent (expected) catalytic activity for ORR and OER. Alloying can integrate the functions of individual components and offer the possibility of generating new functions [10,11]. Transition metal-based alloys have the benefit of high conductivity. However, the alloyed metal nanoparticles also have the shortcomings of poor stability and significant self-aggregation, especially in the harsh electrochemical environment during the process of ORR and oxygen evolution reaction (OER) [12]. Accompanied with the application of metal-based catalysts, carbon is widely adopted as a support for metal catalysts because of its excellent electrical conductivity, appropriate hydrophilicity and high chemical stability. With the development of novel structure carbon nanomaterials, the carbon materials show great promise as a high-performance catalyst for ORR because of their large specific surface area, diverse structure and other characteristics [13,14]. In all the carbon-based materials that have been developed, reduced graphene oxide (RGO) has attracted great attention due to its characteristics of high stability, large specific area, flexible surface modification and low current resistance. Until now, RGO has been adopted in the field of catalysts, novel energy, the chemical industry, etc.

Metal-organic framework (MOF) material refers to a crystalline porous material with a periodic network structure that forms by self-assembly of transition metal ions and organic ligands [15,16]. MOF structure has the advantages of high porosity, low density, large specific surface area, pore rule, adjustable pore size, and topological diversity and structure tailorability, which is considered as an ideal precursor for the synthesis of high-efficiency, metal nitrogen-doped carbon (M–N–C)-based electrocatalysts for ORR [17,18]. Carbon-based materials derived from MOFs, that have the characteristics of highly stable, highly conductive, and porous structures with large specific surface areas, are often employed to prepare nitrogen-doped metal nanoparticle support [19]. Heteroatom-doping (such as nitrogen or boron) has been proved to be an effective strategy for improving materials’ conductivity and electrochemical reaction activity through the structure modulation at the atomic level [20]. Heteroatoms are doped into the sp^2^ lattice of graphitic carbon, which changes the local electronic structure of the matrix carbon. The doped non-metallic heteroatoms usually have different radii and electronegativity, which can activate the charge, and spin density distribution and electronic properties of surrounding carbon in the matrix [21,22]. M–N–C catalysts have been extensively studied due to their unique advantages, such as high efficiency, strong activity, uniform active sites, high atomic efficiency and intrinsic activity. Over the past decades, considerable progress has been made in the M–N–C based catalysts for ORR [23]. The M–N–C structure provides sufficient active sites for oxygen adsorption and reduction, and thus exhibits excellent ORR catalytic activity. As can be seen from the volcanic curve of non-precious metal catalysis for ORR, the Fe–N–C catalyst is currently the most ideal alternative to precious metal catalysts. The presence of metal-nitrogen (M–N) active sites is confirmed by the Mosbauer spectroscopy that the study obtained, which reveals that the Fe atoms with high spin state are considerably active for ORR [24,25,26]. However, the Fe–O bond binding energy is so strong that the desorption of the reaction product from the catalyst surface is inhibited. To further improve the catalytic performance, a secondary metal element could be introduced to regulate the energy level of the Fe atom’s *d* orbital, which could eventually improve the redox potential, weaken the Fe–O bond binding energy and enhance the ORR catalytic performance [27]. FeCo alloy nanoparticles supported by the derivatives of pyrolyzed MOF form stable ORR catalysts with enhanced catalytic activity due to the synergistic effect between the metals and the carbon-doped heteroatoms. However, the uncertainties of MOF pyrolysis often lead to the aggregation of metal particles, low content of doped-N and collapse of the matrix MOF structure [28].

To overcome the shortage of the pyrolysis method, controlling the composition and microstructure of the MOF precursors are effective ways to prepare high-performance electrocatalysts. To improve the conductivity of the catalysts, we have adopted RGO as a support for MOF to synthesize a high-performance catalyst. In this work, we synthesized three kinds of binary FeCo alloy nanoparticles with different atomic ratios of Fe and Co by a facile method and examined their structure and electrocatalytic performance for ORR. Results show that the RGO-supported FeCo alloys have great potential to be high-performance catalysts for ORR. This paper provides a novel strategy to design low-cost, high-performance catalysts for clean energy.

## 2. Experiments

### 2.1. Preparation of Materials

A graphene oxide (GO) solution with the density of 6 g/L was prepared in our laboratory; the details for the preparation are provided in the Appendix A. Except for the GO, the reagents were obtained from Aldrich Co. Ltd. Shanghai (Shanghai, China) and adopted without any treatments. To synthesize the N-doped carbon-supported FeCo alloys, 1 mmol FeCl_3_, 1 mmol CoCl_2_, 2 mmol dipyridine, 2 mmol 1,3,5-benzenetricarboxylic acid, 10 mmol urea and 0.8 g glucose were dissolved in 40 mL H_2_O. Then, 1.0 mL GO solution that sonicated in 20 mL H_2_O for 0.5 h was added to the solution. The mixture solution was continuously stirred for 0.5 h, and then transferred to a 100 mL Teflon autoclave and kept at 200 °C for 5 h. The black precipitate thus obtained was collected and thoroughly washed. The precipitate was moved into a porcelain boat and annealed at 800 °C for 2 h, with a temperature elevating rate of 3 °C/min under argon atmosphere. Finally, catalysts were obtained with the Fe and Co ratio of 1:1, which was named FeCo@N–rGO–1. The other two catalysts, with the atomic ratio of Fe to Co, 3:7 and 7:3, were also synthesized through a similar procedure and named FeCo@N–rGO–2 and FeCo@N–rGO–3, respectively.

### 2.2. Materials Characterization

The physical properties of the catalysts were examined using X-ray diffraction (XRD), transmission electron microscopy (TEM), Raman spectroscopy, and X-ray photoelectron spectroscopy (XPS) tests. The XRD patterns were obtained by D/MAX-2500 (Rigaku, Wilmington, NC, USA) at 40 kV and 300 mA with a step size of 0.04°. The scanning speed of XRD was 6°/min^−1^ in the range of 10° to 90°. TEM tests were conducted by a JEM2100-F (Jeol Ltd., Tokyo, Japan) Shot at 200 kV, Raman spectroscopy was collected by a LabRAM HR UV/Vis/NIR (Horiba Jobin Yvon, Paris, Fance) with a laser source of 514 nm. The XPS tests were performed using a Sigma probe (Thermo VG Scientific, Waltham, MA, USA) equipped with a microfocusing monochromator X-ray source. Scanning electron microscopy (SEM) tests accompanied with energy-dispersive spectroscopy (EDS) were conducted by using a JEOL JSM-6701F electron microscope that worked with a voltage of 5 kV and 10 mA current to examine the structure of the catalysts.

### 2.3. Electrochemical Tests

The electrochemical property tests of the catalysts were performed using a CHI 660E (CH Instruments Inc., Shanghai, China) electrochemical workstation with a three-electrode system. The graphic carbon electrode (ALS Co., Wuhan, China), saturated Hg/Hg_2_Cl_2_ electrode and a glassy electrode were adopted as compare electrode, reference electrode and working electrode, respectively. All potentials shown in this article were converted to the scale of a reversible hydrogen electrode (RHE).

For the electrochemical tests, a 5 mg catalyst of each sample was dispersed in a solution of 0.45 mL water and 0.05 mL Nafion with the content of 1 wt% following with sonication of 30 min. Finally, a black ink was obtained. Then, 5 μL of the ink was pipetted onto the glassy carbon electrode with a diameter of 3 mm and naturally dried at room temperature to form a smooth membrane. Cyclic voltammetry (CV) tests were performed in 0.1 M KOH electrolyte purified with argon or oxygen for more than 40 min to form a saturated solution. The CV tests were conducted with a scanning rate of 5 mV s^−1^ from 0.2 V to 1.2 V (vs. RHE). Similarly, the linear sweep voltammetry (LSV) tests were also conducted under the same conditions, except for using the rotating electrode that rotates with a speed of 1600 rpm. The ORR current was obtained by subtracting the results of the argon purification electrolyte from the LSV result of the oxygen purification electrolyte in order to remove the capacitance of the catalyst. The chronoamperometric test was operated at 0.9 V (vs. RHE) to evaluate the long-time running stability of the catalysts.

## 3. Results and Discussion

### 3.1. Materials Characterization

The structure of the catalysts was firstly examined by XRD tests and the results are displayed in Figure 1a.

Figure 1a shows that three peaks at 44.8°, 65.2°, and 82.6° of the XRD of FeCo@N-rGO–1 were clearly observed. Based on the XRD tests, it was indexed that the three peaks of FeCo@N–rGO–1 were corresponding to (110), (200), and (211) facets, respectively, and the indexed phase composition was FeCo alloy with the JCPDS No. 49-1567. Similarly, based on the XRD test data of t FeCo@N–rGO–2, it was indexed that the peaks at 45.0°, 65.5°, and 85.0° were attributed to the (110), (200), and (211) of Fe_3_Co_7_ alloy with the JCPDS No. 50-0795. Finally, the blue diffraction in Figure 1a shows that the diffraction peaks of FeCo@N–rGO–3 peaks at 44.5°, 64.8°, and 82.2° were attributed to the (110), (200), and (211) facets of the Fe_7_Co_3_ alloy with the indexed JCPDS No. 48-1817. XRD test results confirmed the three alloys were successfully synthesized. It is worth noting that the peak of CoFe_2_O_4_ was also detected in the XRD test results of FeCo@N–rGO–1 and FeCo@N–rGO–3 (the results are supplied in Appendix A), while there is no obvious peak in FeCo@N–rGO–2. This shows that with the increasing percentage of iron element, the precursors of FeCo@N–rGO–1 and FeCo@N–rGO–3 combine oxygen atoms to form CoFe_2_O_4_ during pyrolysis.

Figure 1b shows the Raman spectra of FeCo@N–rGO–1, FeCo@N–rGO–2, and FeCo@N–rGO–3. It clearly shows that in all of three materials, there are two peaks at ~1350 cm^−1^ and ~1580 cm^−1^ corresponding to the D and G bands of the carbon material, which could be attributed to the reduced graphene oxide (rGO). The D band is usually related to the disorder of carbon materials, and the G band is related to the graphitic carbon [29]. The ratio of I_D_/I_G_ reflects the degree of defect of the carbonized material. The I_D_/I_G_ value of FeCo@N–rGO–1 was 1.06, which is higher than that of FeCo@N–rGO–2 (1.01) and FeCo@N–rGO–3 (1.05), indicating a higher degree defective of FeCo@N–rGO–1.

The chemical composition and elemental binding state of FeCo@N–rGO–1 were examined using X-ray photoelectron spectroscopy (XPS) tests, and the results are shown in Figure 1c–f. The full spectra XPS of FeCo@N–rGO–1 is shown in Appendix A. It can be seen that C, N, O, Fe and Co elements coexist in the catalyst. The high-resolution Co 2p spectrum shown in Figure 1c is deconvoluted into three peaks with the combining energy of 779.9, 780.8, and 783.2 eV, which can be attributed to the presence of the metals Co, Co^2+^, and Co–N, respectively [30,31]. The Co species exists in the states of the metal cobalt and Co–N. Similarly, the deconvoluted XPS of Fe 2p in Figure 1d shows the presence of Fe, Fe^2+^, and Fe^3+^ [30,31,32]. The signals of Fe and Co come from the FeCo alloys, while Co^2+^ and Fe^3+^ come from metal oxides, such as CoFe_2_O_4_. The peaks of Fe and Co have different degrees of offset than those of single metals, which may be due to the introduction of N and O elements to adjust the electronic structure [33]. The rGO-supported FeCo alloy promotes the forming of an electron-rich site on the carbon support and an electron-deficient FeCo alloy, which increases the adsorption energy of H and O intermediates. The structure is beneficial for improving the intrinsic activity of the catalytic for ORR [34]. The high-resolution N 1s spectra in Figure 1e are mainly deconvoluted into four peaks with the combining energy of 398.3, 399.4, 400.6, and 401.5 eV, which can be assigned to pyridine-N, metal-N, and pyrrole-N, and graphite-N, respectively [19,35]. Hung and colleagues obtained FeNC/BP by pyrolysis of ammonia at 800 °C, and the XRD and EXAFS test results showed that only Fe_2_N and Fe_2_(N, C) structures were in the material, but we obtained pyridine nitrogen, pyrrole nitrogen and metal nitrogen in an atmosphere of Ar at 800 °C as the result of XPS also confirmed the successful formation of metal nitrides [36]. Pyrrole nitrogen species are highly active sites for nitrogen-doped carbon materials. Pyrrole N and pyridine N species provide anchored sites for central Fe atoms to form active sites for ORR. The graphite N embedded in a carbon matrix can enhance the intrinsic ORR activity in terms of electroconductivity [37,38,39]. It is worth noting that although there are no obvious diffraction peaks observed in the XRD pattern, the presence of metal nitrides can be analyzed from XPS, which could be due to the relative content of metal nitrides compared to the alloy phases that are too small to be detected by XRD test [13]. XPS tests of FeCo@N–rGO–2, FeCo@N–rGO–3 showed there were no peaks of metal nitrides to be detected, which suggests that the composition of the material can be adjusted by changing the molar ratio of the metal precursors [40]. The high-resolution N 1s spectra of FeCo@N–rGO–2 and FeCo@N–rGO–3 are shown in Appendix A, which indicates that there is no M–N signal to be obtained in both catalysts. In addition, the high-resolution O 1s spectra shown in Figure 1f are deconvoluted into four peaks at 530.0, 531.4, 532.4, and 533.2 eV, which could be attributed to M–O–M, C=O, C–O, and COO, respectively [41,42]. Density function theory (DFT) calculations showed that the O_2_ molecules were inherently favored for reduction to H_2_O on N, O co-doped graphene through the direct 4e^−^ pathway [41,43].

To further investigate their structure, the morphology of the catalysts was examined using scanning electron microscopy (SEM) tests and the results are displayed in Figure 2.

Figure 2a shows that the FeCo@N–rGO–1 catalyst is composed of graphene-coated nanoparticles; the diameter of the nanoparticles is about 300~600 nm. Figure 2b,c show the SEM of FeCo@N–rGO–2 and FeCo@N–rGO–3, which clearly reveal that the catalysts were composed by aggregated nanosheet. A similar structure was also observed in the research on Fe/N/C based catalysts [22]. The formation of nanosheets was determined by the pyrolysis of the precursor. On the other hand, rGO also played the role of catalytic support, which could promote the formation of nanosheet.

Figure 2d shows the morphology of FeCo@N–rGO–1, which showed there were some nanosheets fixed on the surface. The corresponded elemental mapping showed that the distribution of Fe, Co, N, O and C were consistent with each other, which proved the successful synthesis of the FeCo based catalysts. On the other hand, the distribution of nitrogen was also consistent with that of carbon, which proved the successful doping of nitrogen to carbon. The elemental mapping also clearly proved the existence of oxygen in the catalyst. This can be caused for two reasons. The first reason is the adsorption of oxygen from the test environment. The nano-sized particles have a strong affinity to oxygen. The second reason for the high content of oxygen could be partial oxidization. The surface of the nano-particle is easily oxidized and formed a CoFe_2_O_4_ based oxide. However, the content is too small to be detected by XRD and other tests.

To further investigate the structure of FeCo@N–rGO–1, transmission electron microscopy (TEM) and high-resolution TEM (HRTEM) tests were conducted; the results are listed in Figure 3.

As can be seen from Figure 3a, the FeCo nanoparticles are uniformly dispersed on the matrix derived from pyrolyzed MOFs and supported by an rGO nanosheet. The self-assembled transition metals and organic ligands formed well-distributed MOFs that had been reported [44]. The formation of the evenly distributed MOFs could be attributed to several factors, such as, precursors of oxygen-rich groups, low iron content, high nitrogen content and high surface area carbon carriers, etc. [45]. Figure 3b shows the structure of the catalysts, that looked spherical. This could be formed during the process of pyrolysis of the MOFs precursor. Based on the TEM tests, it is also observed that the size of FeCo catalysts seems above 200 nm. This can be attributed to the aggregation of FeCo particles and carbon encapsulation. Figure 3b also shows that the nano-carbon layers cover the surface of the FeCo particles, which eventually give the particles a large size.

Figure 3c shows the HRTEM test results for FeCo@N–rGO–1. Two crystal lattices were detected by the HRTEM tests. The crystal lattice of 0.202 nm could be attributed to the (110) facet of the FeCo alloy nanoparticles, which clearly proved the successful synthesis of the FeCo alloy. In addition, the crystal lattice of 0.33 nm could be attributed to the (002) facets of carbon, which was derived from pyrolyzed MOFs [46]. The HRTEM test results also showed that the flake structure of rGO was the support for the FeCo alloy.

### 3.2. Electrochemical Tests

The electrocatalytic performance of the catalyst were firstly examined by cyclic voltammetry (CV) tests in argon- and oxygen-saturated 0.1 M KOH solution. The CV test results of FeCo@N–rGO–1 are shown in Figure 4a. Figure 4a clearly shows that there was no obvious peak detected in the Ar-saturated 0.1 M KOH, which confirmed the high stability of FeCo@N–rGO–1 in alkaline electrolyte. As a contrast, an obvious peak of ORR was detected, which proved the high catalytic activity of FeCo@N–rGO–1 for ORR. CV tests of FeCo@N–rGO–1, FeCo@N–rGO–2 and FeCo@N–rGO–3 with different scan rates are shown in Appendix A, respectively. On the other hand, based on the CV tests, the dependence of *I_p_* to *ω*^1/2^ was also provided. It could be seen from Appendix A that the peak current intensity *I_p_* linearly increased with the increase of *ω*^1/2^, which means the ORR on the catalyst was a diffusion-controlled process.

Figure 4b shows the CV tests of FeCo@N–rGO–1, FeCo@N–rGO–2 and FeCo@N–rGO–3 catalyzed ORR in O_2_-saturated 0.1 M KOH electrolyte. Based on the CV tests, it could be obtained that the peak potential for ORR of FeCo@N–rGO–1, FeCo@N–rGO–2 and FeCo@N–rGO–3 were 0.9 V, 0.87 V and 0.84 V, respectively. On the other hand, it is also distinctly observed that the peak current intensity of FeCo@N–rGO–1 catalyzed ORR is the highest among all the three catalysts. The results mean that the well-dispersed FeCo alloy nanoparticles and the highly connected 3D porous carbon framework significantly improve the electrocatalytic performance for ORR. Combining the performance of the three catalysts, the high catalytic performance could be also attributed to the doped nitrogen of carbon and the metal nitrides, which had been proved by XPS and SEM tests. The doped-nitrogen heteroatoms play the role of regulating the local electronic structure of the catalytic site; nitrogen doping is favorable for the formation of asymmetric spins and better-charged structures in carbon, contributing to a better electrocatalytic property [47]. The synergistic effect between metal nanoparticles and the carbon matrix promoted the electron transport and enhanced the catalytic performance for ORR [48].

Figure 4c shows the linear scanning voltammetry (LSV) test results of the three catalysts. It is also revealed that FeCo@N–rGO–1 shows the highest catalytic performance for ORR, which is consistent with the CV test results. Based on the LSV tests, the onset potential of the catalysts catalyzed are also obtained. The onset potential of the FeCo@N–rGO–1 catalyst 0.96 V (vs. RHE) is higher than that of FeCo@N–rGO–2 catalyst 0.92 V (vs. RHE) and FeCo@N–rGO–3 catalyst 0.90 V (vs. RHE). In addition, FeCo@N–rGO–1 exhibits ORR properties that are superior to other non-precious metal-based catalysts. The corresponding onset potential and half-wave potential are provided in Appendix A [49,50,51,52,53].

Figure 4d shows the Tafel tests of FeCo@N–rGO–1, FeCo@N–rGO–2 and FeCo@N–rGO–3, respectively. The corresponding Tafel slope of the ORR are also provided, which displays that the Tafel slopes of FeCo@N–rGO–1, FeCo@N–rGO–2 and FeCo@N–rGO–3 are 316 mV dec^−1^, 388 mV dec^−1^ and 791 mV dec^−1^, respectively. The least Tafel slope value of FeCo@N–rGO–1 also demonstrates a favorable kinetics mechanism towards ORR for FeCo@N–rGO–1 [54].

To further investigate the catalytic reaction process for ORR of the three catalysts, electrochemical impedance spectra (EIS) tests were carried out in the kinetic region (0.93 V vs. RHE), as shown in Figure 4e. The A.C. impedance spectra can be obtained by processing the size order of the electric double layer capacitors in FeCo@N–rGO–1, FeCo@N–rGO–2 and FeCo@N–rGO–3. At high frequency, according to the geometric formula, the smaller the radius of the AC impedance spectral circle, the larger the capacitance of the electric double layer, the larger the active surface area of the catalyst, and the higher the catalytic activity. The EIS test results indicate that the FeCo@N–rGO–1 catalyst has the largest electrochemical surface area in all the catalysts, which is consistent with its electrochemical properties [55,56]. According to the modulated circuit, the value of R_Ω_ and R_ct_ could be calculated and the results are shown in Appendix A. Appendix A also clearly shows that the R_Ω_ value of FeCo@N–rGO–1 was the lowest in all the catalysts; it also ascertained the highest catalytic activity of FeCo@N–rGO–1 in the three catalysts.

The chronoamperametric tests of FeCo@N–rGO–1, FeCo@N–rGO–2 and FeCo@N– rGO–3 were conducted in the O_2_-saturated 0.1 M KOH electrolyte at 0.93 V, 0.89 V, and 0.87 V (relative to RHE), respectively, and the results are shown in Figure 4f. Figure 4f clearly shows that, after 36,000 s of continuous operation, the current retention rate of FeCo@N–rGO–1 is about 99%, which is much better than that of FeCo@N–rGO–2 (92%) and FeCo@N–rGO–3 (88%). This result shows that the running stability of FeCo@N–rGO–1 was much higher than that of FeCo@N–rGO–2 and FeCo@N–rGO–3.

To further investigate the catalyzed kinetics and reaction pathways for ORR of the catalysts, the rotating ring disc RDE and the rotating ring disc electrode (RRDE) tests were conducted in an O_2_-saturated 0.1 M KOH solution and the results are displayed in Figure 5.

Figure 5a shows the RDE tests of FeCo@N–rGO–1, which clearly showed that the current intensity increased with the increasing rotating speeds. This could be attributed to the enhanced oxygen diffusion on the electrode surface. Figure 5b showed the calculated electron transfer number (*n*) according to the Koutecky–Levich (K–L) equations, which are provided in the Appendix A. Figure 5b distinctly reveals that the calculated *n* was major around 4, which meant that the ORR major happened through a four-electron pathway.

Figure 5c shows the RRDE test results of FeCo@N–rGO–1, FeCo@N–rGO–2, FeCo@N –rGO–3 and Pt/C (20%) at the electrode rotating speed of 1600 rpm. It is clearly shown that the current intensity decreased in the sequence of FeCo@N–rGO–1 > FeCo@N–rGO–2 > FeCo@N–rGO–3. On the other hand, it should be noted also that the maximum current intensity of all the catalysts was higher than that of the Pt/C catalyst, which meant that the catalytic activity of the catalysts was quite approachable to that of the Pt/C catalyst. On the basis of RRDE tests, the electron transfer number (*n*) and the corresponding percentage of peroxide of FeCo@N–rGO–1 were also derived and the results are shown in Figure 5d. Figure 5d distinctly proves that the *n* was about 4 and the H_2_O_2_% was about zero, which also meant ORR major happened through a four-electron pathway. This result is consistent with the RDE test results.

## 4. Conclusions

In this paper, the catalytic performance of rGO supported, FeCo alloy-based catalysts that derived from pyrolysis of iron-cobalt binary MOFs were investigated. The structure of FeCo/rGO based catalysts were confirmed by XRD, SEM and TEM tests. Results showed that the FeCo alloy had a higher catalytic performance for ORR compared with that of rGO supported Fe_3_Co_7_ and Fe_7_Co_3_ alloys. The synergistic effect between the FeCo alloy and N-doped rGO promoted the catalytic performance of FeCo/rGO for ORR. Compared to other ORR catalysts, the rGO support can improve the catalytic performance synergistically. The pyridine nitrogen and pyrrole nitrogen of rGO, the metal nitrides, provided catalytic active sites for ORR. Both the RDE and RRDE tests showed that the ORR major happened through a four-electron pathway. The in situ pyrolysis-prepared FeCo based catalysts showed high long-time running stability. The results of this paper provide a novel way to design high-performance catalysts for ORR from transition metal-based alloys.

## Figures and Tables

**Figure 1 nanomaterials-13-02735-f001:**
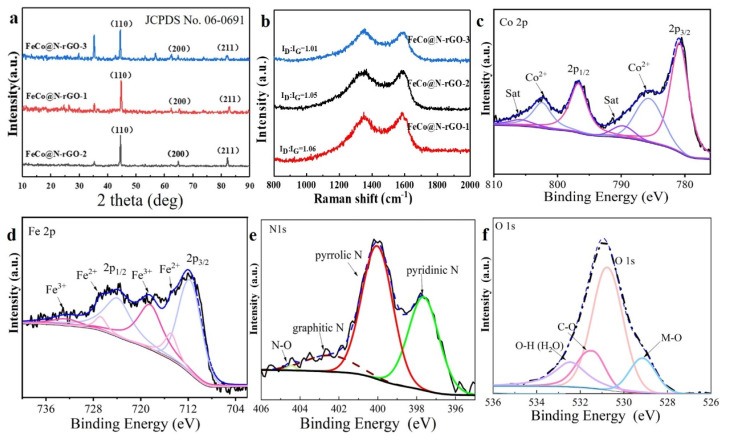
(**a**) XRD patterns and (**b**) Raman spectrum of FeCo@N–rGO–1, FeCo@N–rGO–2 and FeCo@N–rGO–3 annealed at 800 °C under an Ar atmosphere. High-resolution XPS spectra of (**c**) Co 2p (From left to right the curve with purple, grey, pink, purple, grey and pink color are corresponding to deconvoluted satellite line, Co^2+^, Co 2p^1/2^, satellite, Co^2+^, Co 2p^3/2^, respectively), (**d**) Fe 2p (From left to right the curve with purple, pink, grey, purple, pink and grey color are corresponding to deconvoluted Fe^3+^, Fe^2+^, Fe 2p^1/2^, Fe^3+^, Fe^2+^, Fe 2p^3/2^, respectively), (**e**) N 1s (From left to right the curve with light yellow, brown, red and green color are corresponding to deconvoluted N–O, graphitic N, pyrrolic N and pyridinic N, respectively) and (**f**) O 1s (From left to right the curve with light purple, pink, yellow and blue color are corresponding to deconvoluted O–H, C–O, O 1s and M–O, respectively) of FeCo@N–rGO–1.

**Figure 2 nanomaterials-13-02735-f002:**
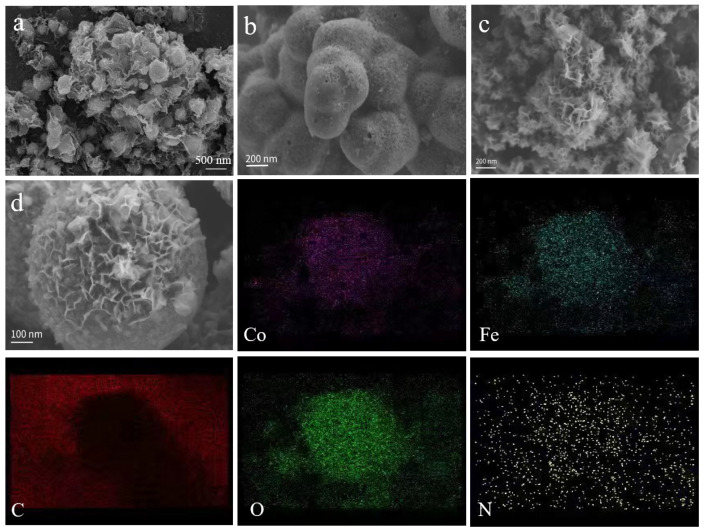
SEM of FeCo@N–rGO–1 (**a**), FeCo@N–rGO–2 (**b**), FeCo@N–rGO–3 (**c**), single particle of FeCo@N–rGO–1 (**d**), and corresponding elemental mapping of Co, Fe, C, O and N.

**Figure 3 nanomaterials-13-02735-f003:**
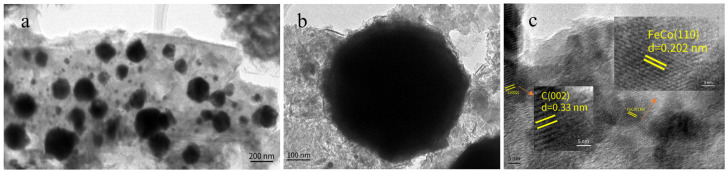
TEM of FeCo@N–rGO–1 (**a**,**b**) and HRTEM of FeCo@N–rGO–1 (**c**).

**Figure 4 nanomaterials-13-02735-f004:**
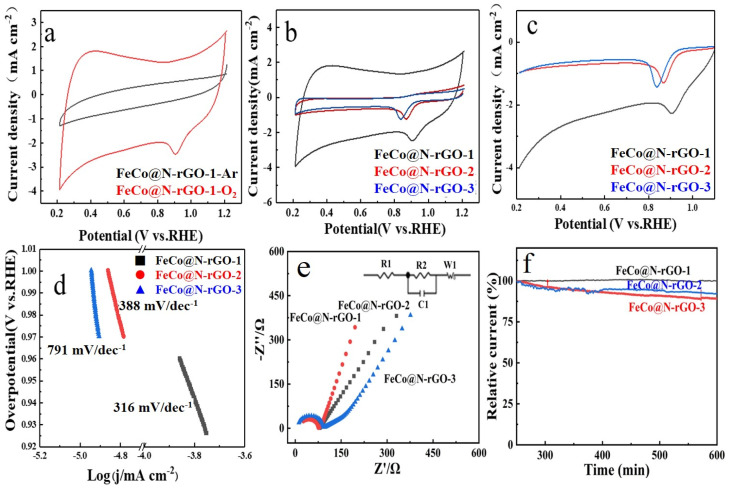
The electrocatalytic activities of FeCo@N–rGO–1, FeCo@N–rGO–2 and FeCo@N–rGO–3. (**a**) ORR Cyclic voltammetry (CV) tests in Ar. (**b**) ORR Cyclic voltammetry (CV) tests in O_2_. (**c**) LSV of FeCo@N–rGO–1, FeCo@N–rGO–2 and FeCo@N–rGO–3. (**d**) Tafel slope of FeCo@N–rGO–1, FeCo@N–rGO–2 and FeCo@N–rGO–3. (**e**) EIS of FeCo@N–rGO–1, FeCo@N–rGO–2 and FeCo@N–rGO–3. (**f**) Chronoamperometric plots of FeCo@N–rGO–1, FeCo@N–rGO–2 and FeCo@N–rGO–3.

**Figure 5 nanomaterials-13-02735-f005:**
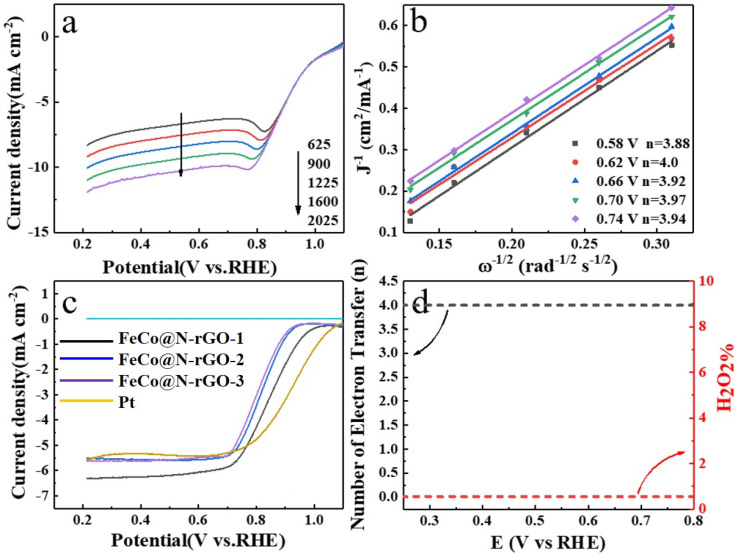
(**a**) RDE polarization curves and corresponding (**b**) Koutecky–Levich (K–L) plots of FeCo@N–rGO–1 in catalyzed ORR in O_2_-saturated 0.1 M KOH with the rotating speed of 1600 rpm. (**c**) RRDE tests of FeCo@N–rGO–1, FeCo@N–rGO–2, FeCo@N–rGO–3 and Pt/C (20%) at 1600 rpm. (**d**) Peroxide percentage and electron transfer number (*n*) of FeCo@N–rGO–1 catalyzed ORR in the potential range of 0.2 V and 0.8 V (vs. RHE).

## Data Availability

Data available on request.

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
