# Peer review of "Reduced Graphene Oxide-Supported Iron-Cobalt Alloys as High-Performance Catalysts for Oxygen Reduction Reaction"

_nanomaterials, 2023, doi:10.3390/nano13192735_

Round 1
Reviewer 1 Report
The article “Reduced graphene oxide supported iron-cobalt alloys as high-performance catalysts for oxygen reduction reaction” by Jun Dong, Shanshan Wang et al is devoted to the synthesis of CoFe particles on the reduced graphene oxide surface and study of their catalytic activities in the ORR. The study is interesting and deserves to be published in the Nanomaterials journal, but after a significant revision. The reviewer's comments are given below.
Authors should correct English in some places. For example, there is an error in the Conclusions in the word "aaloy".
In the Abstract, there are incomprehensible abbreviations; also, the relevance of the study should be highlighted.
The Introduction describes the advantages of MOF, but does not say a word about the use of reduced graphene oxide. The authors should separate the concepts of MOF and reduced graphene oxide. Their system does not apply to the MOF. Nanoparticles are located on the RGO surface without a rigid framework structure.
From the phrase “So, in this work, we synthesized three kinds of binary FeCo alloy nanoparticles with different atomic proportion by a facile method and tested their structural and electrocatalytic performance for ORR”, we can conclude that only pure nanoparticles were used for catalysis. It is necessary to correct text of the article according to the study.
In the Experimental part, authors should replace the designations of hydrates (FeCl3-6H2O and CoCl2-6H2O). SEM and EDX is not mentioned in the list of equipment.
It is necessary to separate the analyzes presented in Fig. 1. XRD is very difficult to correlate with the text. But this is the main proof of real phase composition of the prepared nanoparticles. Signatures are too small; it will be difficult for the reader to read.
The difference in the ID/IG ratio is completely insignificant and is within the limits of the instrumental calculation error to say that one sample has a higher degree of defectiveness than the others.
From the presented SEM image, it is not clear why so much oxygen atoms are concentrated in the nanoparticle surface. Obviously, an oxygen can be present in the structure of reduced graphene oxide, although not much of it should remain after calcining the samples at 800°C in an inert atmosphere. Thus, the alloy definitely contains iron-cobalt oxide.
When describing TEM, the authors say “which was derived from the rGO support and the MOFs pyrolysis derived carbon [43]”. What is meant by this phrase? The experiment indicated that, in addition to salt precursors, a large amount of organic substances was mixed with the GO dispersion. They were supposed to give the same MOF system? That is, the samples contain different carbon - from RGO and from residual organic matter? But most likely there will be amorphous carbon from organics. Authors should pay more attention to the identification of materials.
The phrase "The synergistic effect between the alloys and N-doped rGO promoted the catalytic performance of FeCo/rGO for ORR" does not correlate with the fact that FeCo/rGO has a higher catalytic activity than the other two samples with the same composition. One could speak of a synergistic improvement when compared to other ORR catalysts.
Authors should correct English in some places. For example, there is an error in the Conclusions in the word "aaloy".
Author Response
Reviewer #1
The article “Reduced graphene oxide supported iron-cobalt alloys as high-performance catalysts for oxygen reduction reaction” by Jun Dong, Shanshan Wang et al is devoted to the synthesis of CoFe particles on the reduced graphene oxide surface and study of their catalytic activities in the ORR. The study is interesting and deserves to be published in the Nanomaterials journal, but after a significant revision. The reviewer's comments are given below.
Reviewer 1:Authors should correct English in some places. For example, there is an error in the Conclusions in the word "aaloy".
Response: Thanks for the reviewer’s careful examination. The typo has been corrected.
Revision: Page 15 Line 6:
In this paper, the catalytic performance of rGO supported FeCo alloy based catalysts that derived from pyrolysis of iron-cobalt binary MOFs were investigated.
Reviewer 2:In the Abstract, there are incomprehensible abbreviations; also, the relevance of the study should be highlighted.
Response: Thanks for the reviewer’s constructive comments. We have re-edited the Abstract.
Revision: Page 1 Abstract:
Exploring non-precious metal-based catalysts for oxygen reduction reactions (ORR) to substitute the precious metal catalysts has attracted great attention at present. In this paper, we report a general methodology to prepare nitrogen-doped reduced graphene oxide (N-rGO) supported FeCo alloy (FeCo@N-rGO) based catalysts for ORR. The structure of the FeCo@N-rGO based catalysts is investiagted by X-ray diffraction, scanning electron microscopy, X-ray photoelectron spectroscopy, and transition electron microscopy, etc. Results show that the FeCo alloy is supported by the rGO and carbon that derived from the organic ligand of Fe and Co ions. The eletrocatalytic performance is examined by cyclic voltammetry, linear scanning voltammetry, Tafel, electrochemical spectroscopy impedance, rotate disc electrode, and rotate ring disc electeode, etc. Results show that FeCo@N-rGO based catalysts exhibit an onset potential of 0.98 V (vs. RHE) and half-wave potential of 0.93 V (vs. RHE). The excellent catalytic performance of FeCo@N-rGO is assigned to the large surface area and the synergistic effect between FeCo alloy and N-rGO, which provides a large number of active sites and sufficient surface area.
Reviewer 3:The Introduction describes the advantages of MOF, but does not say a word about the use of reduced graphene oxide. The authors should separate the concepts of MOF and reduced graphene oxide. Their system does not apply to the MOF. Nanoparticles are located on the RGO surface without a rigid framework structure.
Response: Thanks for the reviewer’s careful examination. We supplied some introduction of RGO.
Revision: Page 3 Line 1:
In all the developed carbon based materials, reduced graphene oxide (RGO) have attracted great attention due to its characteristics of high stability, large specific area, flexible surface modification and low current resistensity. Until now, the RGO have been adopted in the field of catalysts, novel energy, chemical industry and etc.
Page 4 Line 13:
To improve the conductivity of the catalysts, we adopt RGO as support for MOF to synthesize high perormance catalyst.
Reviewer 4:From the phrase “So, in this work, we synthesized three kinds of binary FeCo alloy nanoparticles with different atomic proportion by a facile method and tested their structural and electrocatalytic performance for ORR”, we can conclude that only pure nanoparticles were used for catalysis. It is necessary to correct text of the article according to the study.
Response: Thanks for the reviewer’s constructive comments. We have revised this part in the paper.
Revision: Page 4 Line 14:
In this work, we synthesized three kinds of binary FeCo alloy nanoparticles with different atomic ratio of Fe and Co by a facile method and examined their structure and electrocatalytic performance for ORR.
Reviewer 5:In the Experimental part, authors should replace the designations of hydrates (FeCl3-6H2O and CoCl2-6H2O). SEM and EDX is not mentioned in the list of equipment.
Response: Thanks for the reviewer’s careful examination. We have supplied the equipments for SEM and EDX tests.
Revision: Page 4 Line 25:
To synthesize the N-doped carbon supported FeCo alloys, 1 mmol FeCl3, 1 mmol CoCl2, 2 mmol dipyridine, 2 mmol 1,3,5-benzenetricarboxylic acid, 10 mmol urea and 0.8 g glucose were dissolved in 40 mL H2O.
Page 1 Line 16:
Scanning electron microscopy (SEM) tests accompanied with energy dispersive spectroscopy (EDS) were conducted by using a JEOL JSM-6701F electron microscope that worked with the voltage of 5 kV and 10 mA current to examine the structure of the catalysts.
Reviewer 6:It is necessary to separate the analyzes presented in Fig. 1. XRD is very difficult to correlate with the text. But this is the main proof of real phase composition of the prepared nanoparticles. Signatures are too small; it will be difficult for the reader to read.
Response: Thanks for the reviewer’s careful examination. We have revised the signature in Fig. 1.
Revision: Page 6 Fig. 1
Fig. 1 (a) XRD patterns and (b) Raman spectrum of FeCo@N-rGO-1, FeCo@N-rGO-2 and FeCo@N-rGO-3 annealed at 800 ℃ under an Ar atmosphere. High-resolution XPS spectra of (c) Co 2p, (d) Fe 2p, (e) N 1s and (f) O 1s of FeCo@N-rGO-1.
Reviewer 7:The difference in the ID/IG ratio is completely insignificant and is within the limits of the instrumental calculation error to say that one sample has a higher degree of defectiveness than the others.
Response: Thanks for the reviewer’s careful examination. The values of ID/IG ratio is tightly related with the structure defects. The three kinds of FeCo based catalyst are maken under same condiction, which make the structure of derived cabon and RGO are quite similar. So, the detected ID/IG ratio values ratio are quite approach and similar.
Reviewer 8:From the presented SEM image, it is not clear why so much oxygen atoms are concentrated in the nanoparticle surface. Obviously, an oxygen can be present in the structure of reduced graphene oxide, although not much of it should remain after calcining the samples at 800°C in an inert atmosphere. Thus, the alloy definitely contains iron-cobalt oxide.
Response: Thanks for the reviewer’s careful examination. The content of oxygen should be attributed to the absorbed oxygen from test environment. The FeCo alloy particles is so small that the oxygen can be easily absorbed during the process of prepare samples for SEM tests.
Reviewer 9:When describing TEM, the authors say “which was derived from the rGO support and the MOFs pyrolysis derived carbon [43]”. What is meant by this phrase? The experiment indicated that, in addition to salt precursors, a large amount of organic substances was mixed with the GO dispersion. They were supposed to give the same MOF system? That is, the samples contain different carbon - from RGO and from residual organic matter? But most likely there will be amorphous carbon from organics. Authors should pay more attention to the identification of materials.
Response: Thanks for the reviewer’s constructive comments. We have revised the color of the materials.
Revision: Page 10 Line 21:
In addition, the crystal lattice of 0.33 nm should be attributed to the (002) facets of carbon, which was derived from pyrolyzed MOFs [46].
Reviewer 10:The phrase "The synergistic effect between the alloys and N-doped rGO promoted the catalytic performance of FeCo/rGO for ORR" does not correlate with the fact that FeCo/rGO has a higher catalytic activity than the other two samples with the same composition. One could speak of a synergistic improvement when compared to other ORR catalysts.
Response: Thanks for the reviewer’s constructive comments. We have revised the color of the materials.
Revision: Page 11 Line 24:
The synergistic effect between the FeCo alloy and N-doped rGO promoted the catalytic performance of FeCo/rGO for ORR. Compared to other ORR catalysts, the rGO support can improve the catalytic performance synergistically.

Reviewer 2 Report
In this manuscript, Dong et al. reported the synthesis of N-doped reduced graphene oxide supported iron-cobalt alloys via pyrolysis of MOF precursors and their application as high-performance catalysts for oxygen reduction reaction (ORR). In particular, the atomic proportions of FeCo alloys were optimised to deliver the best ORR performance. Overall, this research work has acceptable novelty and the manuscript was generally well presented. This work is a good fit for the journal Nanomaterials. However, some technical issues need to be resolved before the possible acceptance of the manuscript. Please see below for more detail.
1. Figure 1a and 1b, the color codes for different samples are not consistent, which can be confusing. Please revise to improve clarity.
2. Figure 1c and 1d, the analysis of XPS data needs revision. For any species, it should have one set of peaks, i.e., 2p3/2 and 2p1/2. In this case, Co-Nx does not have a 2p1/2 peak identified, and Fe2+, Fe0, and Fe-Nx only have one peak each.
3. Recent works on ORR catalysts (e.g., Small Methods, 2018, 2, 1800071; Energy Fuels 2021, 35, 13585) are suggested to be included in Introduction.
4. There is an inconsistency in the binding energy position of peaks. For Co XPS data, metallic Co (Co0) was sitting at a lower binding energy than Co2+, however, for the Fe XPS data, Fe0 species was sitting at a higher binding energy than Fe2+ and Fe3+.
5. Some of the content was repeated in the Supporting Information, such as experimental detail, XRD figure, etc.
6. MOF materials are indeed a good platform for making catalysts. Related works (e.g., SusMat, 2021, 1, 460-481) can be referenced in Introduction.
7. Figure 4c, why the limiting current density was so low? Did the authors apply any rotation during the recording of the LSV polarization data?
8. Figure 5d, why were the electron transfer number and H2O2% almost a straight line without any change?
9. Is N-doping also important to the good catalytic performance? The authors highlighted this point in the Abstract, which, however, was not reflected in the Title.
Minor editing of English language is required. For instance, "3D" in the second paragraph of the Introduction should be revised into "3d".
Author Response
Reviewer #2
In this manuscript, Dong et al. reported the synthesis of N-doped reduced graphene oxide supported iron-cobalt alloys via pyrolysis of MOF precursors and their application as high-performance catalysts for oxygen reduction reaction (ORR). In particular, the atomic proportions of FeCo alloys were optimised to deliver the best ORR performance. Overall, this research work has acceptable novelty and the manuscript was generally well presented. This work is a good fit for the journal Nanomaterials. However, some technical issues need to be resolved before the possible acceptance of the manuscript. Please see below for more detail.
Reviewer: 1. Figure 1a and 1b, the color codes for different samples are not consistent, which can be confusing. Please revise to improve clarity.
Response: Thanks for the reviewer’s constructive comments. We have revised the color of the materials.
Revision: Page 6 Fig. 1
Reviewer: 2. Figure 1c and 1d, the analysis of XPS data needs revision. For any species, it should have one set of peaks, i.e., 2p3/2 and 2p1/2. In this case, Co-Nx does not have a 2p1/2 peak identified, and Fe2+, Fe0, and Fe-Nx only have one peak each.
Response: Thanks for the reviewer’s careful examination and constructive comments. We have revised the deconvoluted results according to your comments.
Revision: Page 6 Line 1:
Reviewer: 3. Recent works on ORR catalysts (e.g., Small Methods, 2018, 2, 1800071; Energy Fuels 2021, 35, 13585) are suggested to be included in Introduction.
Response: Thanks for the reviewer’s constructive comments. It could be seen that the reviewer is an expert in this field. We have compared the our results to the mentioned reference.
Revision: Page 2 Line 17, references 8 and 9:
In theory, the transition metal-based catalysts have great potential to be high performance catalysts for oxygen reduction and other energy storage [8, 9].
[8] X. Xu, W. Wang, W. Zhou, Z. Shao, Recent Advances in Novel Nanostructuring Methods of Perovskite Electrocatalysts for Energy-Related Applications, Small Methods, 2 (2018) 1800071.
[9] X. Xu, C. Su, Z. Shao, Fundamental Understanding and Application of Ba0.5Sr0.5Co0.8Fe0.2O3−δ Perovskite in Energy Storage and Conversion: Past, Present, and Future, Energy & Fuels, 35 (2021) 13585-13609.
Reviewer: 4. There is an inconsistency in the binding energy position of peaks. For Co XPS data, metallic Co (Co0) was sitting at a lower binding energy than Co2+, however, for the Fe XPS data, Fe0 species was sitting at a higher binding energy than Fe2+ and Fe3+.
Response: Thanks for the reviewer’s careful examination and constructive comments. We have revised the deconvoluted results according to your comments.
Revision: Page 6 Fig 1:
Reviewer: 5. Some of the content was repeated in the Supporting Information, such as experimental detail, XRD figure, etc.
Response: Thanks for the reviewer’s careful examination. We have reedited the XRD of the Supporting Information. The XRD in Supporting Information major illutrates the formation of micro-amount metal oxide, such as CoFe2O4.
Revision: Supporting Information Fig. S1.
Fig. S1 (a) XRD patterns of FeCo@N-rGO-1, FeCo@N-rGO-2 and FeCo@N-rGO-3. Full spectra XPS of FeCo@N-rGO-1 (b), High resolution N 1s (c) and Full spectra XPS (d) of FeCo@N-rGO-2; High resolution N 1s (e) and full spectra XPS (f) of FeCo@N-rGO-3.
Reviewer: 6. MOF materials are indeed a good platform for making catalysts. Related works (e.g., SusMat, 2021, 1, 460-481) can be referenced in Introduction.
Response: Thanks for the reviewer’s constructive comments. We have compared our reults with the references. The mentioned reference is cited in the paper as reference [20].
Revision: Page 3 Line 15, Reference 20:
Heteroatoms, such as nitrogen, boron, doping has been proved to be an effective strategy for improving materials conductivity and electrochemical reaction activity through the structure modulation in atomic level [20].
[20] X. Xu, H. Sun, S.P. Jiang, Z. Shao, Modulating metal–organic frameworks for catalyzing acidic oxygen evolution for proton exchange membrane water electrolysis, SusMat, 1 (2021) 460-481.
Reviewer: 7. Figure 4c, why the limiting current density was so low? Did the authors apply any rotation during the recording of the LSV polarization data?
Response: Thanks for the reviewer’s careful examination. The LSV tests were conducted just used the glass-carbon electrode without any rotation. This results in the obtained current rather low. The RDE test results show that the current greatly increased with the rotating electrode.
Reviewer: 8. Figure 5d, why were the electron transfer number and H2O2% almost a straight line without any change?
Response: Thanks for the reviewer’s careful examination. The obtained straight line should be attributed to the selection of potential range. We just adopt the potential range between 0.2 and 0.8 V (vs RHE), in which the current is fairly stable. When the potential above 0.8 V (vs RHE), the electron tansfer number and the percentage of H2O2 have great changes.
Reviewer: 9. Is N-doping also important to the good catalytic performance? The authors highlighted this point in the Abstract, which, however, was not reflected in the Title.
Response: Thanks for the reviewer’s careful examination. The N-doping usually has great influences on the catalytic performance for ORR. Researches on the carbon based catalysts for ORR usually pay attention to heteroatoms doping. But, in this paper, we put the emphasis on FeCo alloy. So, we donot put the N-doping in the title.

Round 2
Reviewer 1 Report
The authors of the article "Reduced graphene oxide supported iron-cobalt alloys as high-performance catalysts for oxygen reduction reaction" took into account some of the reviewer's comments and corrected the text. However, the reviewer still had some comments about the study.
First, fig. 1, 4, 5 and S1 are too small. Signatures are completely invisible, which causes inconvenience for the reader. On fig. S2 signature errors.
Second, the main remark was related to the phase composition of the obtained composites. XRD is the main demonstrative method for confirming the phase composition of the obtained samples, namely iron and cobalt alloys. You should make a larger image of the X-ray diffraction patterns and show that they correspond to the JCPDS references. In its present form, it is not at all clear which reference is shown in purple, and the reference for CoFe2O4 is generally in SI. In combination with SEM, it is obvious that a large percentage of the oxide phase is formed and this is not due to sample preparation for microscopy. If sample preparation for SEM is poor, then can such scientific results be used at all?
Round 3
Reviewer 1 Report
The authors' responses do not satisfy the reviewer's previous comments. There are still comments about the images. Signatures in Fig. 1 and 4 are too small. The reader will not even be able to see the labels and axis designations.
Next, after removing the XRD diffraction pattern for Fe2CoO4, it is worth returning to the question of why there is so much oxygen in the EDX image (Fig. 2). The authors logically explained this fact in previous versions; they should only have corrected the XRD and not removed it completely.
The question that arose during the first review also arises again. Based on particle sizes, can FeCo particles up to 200 nm be considered as nanoparticles? It should also be noted in the article that using the method produces a very wide size distribution.
Author Response
Reviewer #1
The authors' responses do not satisfy the reviewer's previous comments. There are still comments about the images. Signatures in Fig. 1 and 4 are too small. The reader will not even be able to see the labels and axis designations.
Response:Thanks for the reviewer’s patient for the review to our paper. Now, we have revised Fig. 1 and 4 according to reviewer’s constructive comments.
Revision: Page 6 Fig. 1
Fig. 1 (a) XRD patterns and (b) Raman spectrum of FeCo@N-rGO-1, FeCo@N-rGO-2 and FeCo@N-rGO-3 annealed at 800 ℃ under an Ar atmosphere. High-resolution XPS spectra of (c) Co 2p, (d) Fe 2p, (e) N 1s and (f) O 1s of FeCo@N-rGO-1.
Page 12 Fig. 4
Fig. 4 The electrocatalytic activities of FeCo@N-rGO-1, FeCo@N-rGO-2 and FeCo@N-rGO-3. (a) ORR Cyclic voltammetry (CV) tests in Ar. (b) ORR Cyclic voltammetry (CV) tests in O2. (c) LSV of FeCo@N-rGO-1, FeCo@N-rGO-2 and FeCo@N-rGO-3. (d) Tafel slope of FeCo@N-rGO-1, FeCo@N-rGO-2 and FeCo@N-rGO-3. (e) EIS of FeCo@N-rGO-1, FeCo@N-rGO-2 and FeCo@N-rGO-3. (h) chronoamperometric plots of FeCo@N-rGO-1, FeCo@N-rGO-2 and FeCo@N-rGO-3.
Reviewer #2
Next, after removing the XRD diffraction pattern for Fe2CoO4, it is worth returning to the question of why there is so much oxygen in the EDX image (Fig. 2). The authors logically explained this fact in previous versions; they should only have corrected the XRD and not removed it completely.
Response: Thanks for the reviewer’s constructive comments. We explained the reason for high oxygen content in text.
Revision: Page 10 Line 3
The elemental mapping also clearly proved the existence of oxygen in the catalyst. This should be caused by two reasons. The first reason is the adsorption of oxygen from test environment. The nano-sized particles have strong affinity to oxygen. The second reason for the high content of oxygen should be partial oxidization. The surface of the nano-particle is easily to oxidized and forming CoFe2O4 based oxide. But, the content is too less to be detected by XRD and other tests.
Reviewer #3
The question that arose during the first review also arises again. Based on particle sizes, can FeCo particles up to 200 nm be considered as nanoparticles? It should also be noted in the article that using the method produces a very wide size distribution.
Response: Thanks for the reviewer’s careful examination. Despite the particle size seems up to 200 nm, the particle is encapsulated by surface carbon layer. On the other hand, the FeCo particle size have some aggregation during the preparation process. All the factors make the particles shown a large size. In fact, the FeCo particle are in nano-size.
Revision: Page 10 Line 22:
Based on the TEM tests, it is also observed that the size of FeCo catalysts seems above 200 nm. This can be attributed to the aggregation of FeCo particles and the carbon encapsulation. Fig. 3b also shows the nano-carbon layers covers the surface of the FeCo particles, which eventually make the particles show a large size.
